# Evaluation of the Soothing and Protective Properties of a Lignin Hydrolyzate

**Letteria Greco [1,*], Salviana Ullo [2,*], Luigi Rigano [3], Marco Fontana [1] and Enzo Berardesca [4]**

1   Sinerga Group, Via della Pacciarna 67, 21050 Gorla Maggiore, (VA), Italy
2   Sinerga SPA, Via della Pacciarna 67, 21050 Gorla Maggiore, (VA), Italy
3   Rigano Laboratories Srl, 20125 Milan, Italy
4   Istituto Dermatologico S. Maria e S. Gallicano, IRRCS Via Chianesi 53, 00144 Roma, Italy
*   Correspondence: l.greco@sinerga.it (L.G.); s.ullo@sinerga.it (S.U.)

**Abstract:** Lignins have shown remarkable antioxidant properties; acting as "scavengers" of free radicals physiologically produced by cell metabolisms; and exerting a protective action caused by the strong ability of these molecules to absorb UV radiation. Through preliminary Molecular Modeling studies and experimental studies in vivo and in vitro, a lignin hydrolysate compound has been shown to be an extremely versatile active ingredient, presenting soothing, anti-inflammatory, anti-itch, anti-oxidant, anti-aging and anti-pollution properties. The possible fields of application are therefore multiple; making this lignin hydrolysate a particularly interesting ingredient for topical dermatological compositions in the treatment of various skin disorders such as inflammation, edema, swelling, rash, redness, itching, chrono- and photo-induced skin aging. These manifestations are also the basis of more or less serious skin problems, making lignin hydrolysate capable of being used in cosmetic products for the eternal challenge of fighting skin aging, but also in medical devices that can be used to fight more painful and annoying symptoms, like those caused by dermatitis or psoriasis.

**Keywords:** lignin; polyphenols; skin disorders; dermatitis; psoriasis

## 1. Introduction

The objective of the present article is to explore a lignin hydrolyzate extracted from the bark of Pinus Taeda and Pinus Radiata pines originating from North America, which contains oligomers comprising at least one of the monomeric units selected from synapyl alcohol, coniferyl alcohol and coumaryl alcohol, for use in the treatment of skin disorders including: inflammation, edema, swelling, rash, redness, itching, chrono- and photo-induced skin aging and dermatitis. The active ingredient has a polyphenol titer higher than 85% (w/w) in weight on the total weight of lignin hydrolysate, out of which at least 25% is reactive to Folin-Ciocalteu.

The lignin hydrolyzate object of the present article is obtainable by using techniques comprising hydrolysis of the lignocellulose from virgin wood, by removing cellulose and hemicellulose; basic hydrolysis (NaOH), acid reprecipitation ($CO_2$ and $H_2SO_4$) and a washing with water; then followed by filtration and drying under controlled conditions, without using organic solvents for extraction [1]. Hydrolysis reaction results in the separation of a larger molecule into component parts. The structure of hydrolyzed lignin is more condensed in comparison with native lignin [2].

Lignin is one of the main constituents of plants, as it represents 15–35% of the mass, depending on the species. It is located between the cell walls of plant fibers and performs the function of a binder, providing hardness and rigidity to the plant. Chemically, lignin is an irregular three-dimensional biopolymer, with a relatively complex structure, synthesized through the dehydrogenative polymerization of p-cumaryl alcohol, coniferyl and sometimes synapyl alcohol. The

molecular weight of the native lignin has an average value of 20,000 Da, depending on the source and the processes by which the polymer is extracted.

The fundamental monomeric unit can be identified in the phenyl-propane, bound with 4-O—β ether bonds, however, the presence of different functional groups (methyl, ethyl, methoxy, hydroxyl, etc.) makes the lignin of each plant species different from the others [3] (Figure 1).

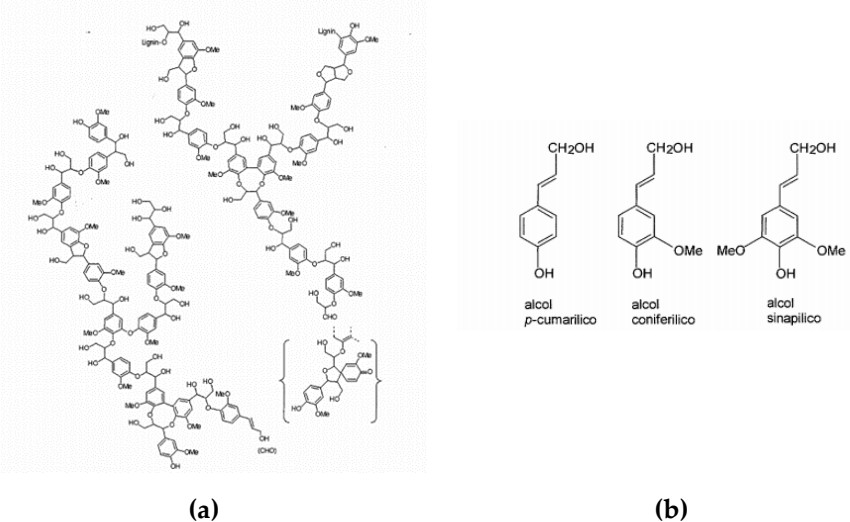

**(a)**                                    **(b)**

**Figure 1.** (**a**) Softwood Lignine structure according to Brunow, Sipila e Syrjanen. (**b**) On the right, structure of the three Lignin's precursors (°p-coumaryl alcohol, coniferyl alcohol, and sinapyl alcohol).

From a chemical-structural point of view, lignin belongs to the class of non-flavonoid polyphenols, secondary metabolites of plants characterized by the presence of multiple phenolic groups associated in more or less complex structures generally of high molecular weight.

Through preliminary studies of Molecular Modelling and experimental studies in vivo and in vitro, it has been found that hydrolyzed lignin has soothing, anti-inflammatory, anti-aging, antioxidant and anti-itch properties that make it a particularly interesting active ingredient for topical or oral compositions for dermatological use. This Molecular Modelling study has shown interesting score values, in particular against Arachidonate 5-lipoxygenase. Arachidonate 5-lipoxygenase is an enzyme involved in the synthesis of leukotrienes from arachidonic acid. Arachidonic acid in fact, released by lipid membranes in response to damage and irritation, is converted by arachidonate 5-lipoxygenase into (6E, 8Z, 11Z, 14Z)-(5S)-5-hydroperoxicose-6,8,11,14-tetrahenoate, precursor of leukotrienes. These leukotrienes are, in turn, related to the production of inflammation mediators, such as histamine and prostaglandins. Given its involvement in the inflammatory response, numerous evidence attributes a key role to arachidonate 5-lipoxygenase for the treatment of various inflammatory diseases of the skin, such as psoriasis and dermatitis.

## 2. Anti-Inflammatory Effect

### 2.1. In Vitro Evaluation of the Anti-Inflammatory Potential on Reconstituted Human Epidermis

Inflammation plays a major role in the induction and progression of several skin diseases. IL-1 is one of the most well-known inflammatory markers. Interleukin-1 plays a key role in inflammation and keratinocyte activation [4]. IL-1 family is a group of 11 cytokines: interleukin (IL) 1 alpha (IL-1α) and 1 beta (IL-1β) isoforms are the most investigated. Indeed the inflammatory process is initiated by IL-1α, but is propagated and maintained by both isoforms.

Overexpression of both interleukines is positively correlated with some skin disease as psoriasis, atopic dermatitis, skin cancer and skin phototoxicity, etc. [5].

Purpose of the Test

The objective of the study was to assess quantitatively the effects of the sample in protecting skin cells and in inhibiting the inflammatory reaction caused by moderate irritating agents as sodium lauryl sulphate (SLS) by means of a multilayer in vitro skin cell model. Indeed, we used SLS as an irritating agent to simulate an inflammatory condition. As a reaction, cells started to produce IL-1α. We investigated this through a specific ELISA assay, the inhibition of the SLS-induced release of IL-1α cytokine following exposure to the tested substance and respect to skin untreated samples. A reconstructed artificial human skin model was made comprising normal human epidermal keratinocytes, growing as an integrated three-dimensional cell culture model, perfectly mimicking the human skin in vitro. The model exhibits normal barrier functions (presence of a well-differentiated stratum corneum). Active ingredient concentrations were chosen after an MTT cell viability assay [6]. The Role of Interleukin-1 in Inflammatory and Malignant Human Skin Diseases and the Rationale for Targeting Interleukin-1 Alpha).

Treatment and Exposure

Pre-treated epidermis for 40 minutes with SLS at 0.3% has been madder in order to induce IL-1α synthesis and release. The complete test was performed following an MTT cell viability assay [6].

Cytokine concentration is determined using a standard curve. IL1-alpha release inhibition has determined as:

%IL1-alpha inhibition: 100- (pg/mL IL-1alpha release by irritated sample/ pg/mL IL-1alpha positive control)*100.

*2.2. Results and Conclusions*

Results Interpretation

Acceptance criteria of the assay complied, hence the assay is valid [6].

Conclusions

On the basis of the results shown here (Table 1), the sample:

- Solution 0.1% sample of active ingredient does show in vitro anti-inflammatory activity. After 24 hours incubation, an IL1alpha release reduction (31.25%) is pointed out.
- Solution 0.3% sample of active ingredient does show in vitro anti-inflammatory activity. After 24 hours incubation, an IL1alpha release reduction (28.77%) is pointed out.

**Table 1.** Results of the inhibition of IL-1α release.

| Sample | IL-1α pg/mL | | | |
| | 6h | | 24h | |
| | pg/mL Mean | % Inhibition | pg/mL Mean | % Inhibition |
|---|---|---|---|---|
| Negative control (Phosphate Buffer -PBS) | 18.05 | | 28.58 | |
| SLS 0.3% (Positive control) | 57.03 | | 294.82 | |
| Solution 0.1% Active Ingredient | 38.93 | | 55.70 | |
| Solution 0.1% + SLS 0.3% | 67.23 | No inhibition | 202.70 | 31.25 |
| Solution 0.3% Active Ingredient | 64.59 | | 76.77 | |
| Solution 0.3% + SLS 0.3% | 43.35 | 23.98 | 144.37 | 28.77 |

## 3. Anti Oxidant Effect

The antioxidant capability of a topical product has been evaluated through an in vitro test (in tube enzymatic reaction). Hydrolyzed lignine has shown an antioxidant effect, thereby increasing the activity of superoxide dismutase (SOD) with consequent antioxidant properties.

Purpose of the Test

A direct measure of the increase of the activity superoxide dismutase (SOD) by the tested product was tested through colorimetric analysis. The purpose of the present test is to investigate the anti-oxidant properties of a raw material or finished product through the determination of the activity

increase of the superoxide dismutase enzyme [7]. The inhibition of a chromogenic substrate, formazan, detectable at 450nm, has been evaluated. The color development is inhibited when any antioxidant agent was added in the reaction. The antioxidant capability of the tested substance was quantitatively measured as inhibition percentage of the color compound related to the negative control (water alone).

The sample of 10% active ingredient has been solubilized in glycerin. So we tested the final following concentrations in water: 0.25%, 0.15% e 0.05%.

SOD activity is calculated as percentage of dye compound reduction (inhibition rate%).

SOD activity (inhibition rate%) = $[(A_{CN} - A_{blank2}) - (A_{sample} - A_{blank1}) / (A_{CN} - A_{blank2})] \times 100$

Results

As shown in Table 2, sample containing cosmetic active ingredient (hydrolyzed lignin) is able to increase SOD activity in a dose-dependent way, so it shows antioxidant activity. The highest effect was pointed out at a 0.25% concentration of the sample, where the inhibition of the formazan dye production was 75%, indicative of an effective increase of the SOD activity.

**Table 2.** SOD Activity Inhibition.

| Sample | OD | Activity SOD (Inhibition Rate%) |
|---|---|---|
| Negative control | 0.2480 | - |
| | 0.2550 | |
| Active Ingredient | 0.1420 | 75.0% |
| Batch: n.p. 0.25% | 0.1440 | |
| Blank 1 | 0.1010 | - |
| Active Ingredient | 0.0860 | |
| Batch: n.p. 0.25% | | |
| Active Ingredient | 0.1490 | 64.4% |
| Batch: n.p. 0.15% | 0.1510 | |
| Blank 1 | 0.0860 | - |
| Active Ingredient | 0.0730 | |
| Lotto/Batch: n.p. 0.15% | | |
| Active Ingredient | 0.1860 | 32.3% |
| Lotto/Batch: n.p. 0.05% | 0.2040 | |
| Blank 1 | 0.0590 | - |
| Active Ingredient | 0.0630 | |
| Batch: n.p. 0.05% | | |
| Blank 2 | 0.0530 | - |
| | 0.0540 | |

In vitro evaluation of the protective effects against the cell release of reactive oxygen species (ROS) after UVA exposure was compared to ascorbic acid.

Purpose of the Test

UVA in a skin-derived cell model. The stimulus with UVA rays is used to trigger the cell oxidative response [8]. The intensity of ROS release was measured through fluorimetric analysis. DCFH-DA is a redox-sensitive fluorescence probe that easily diffuses inside the cells, where it loses its di-acetate unit by cell esterases cleavage. It is converted into DCFH, exposing its reactive site that reacts with intra-cellular ROS and is oxidized into dichlorofluorescin (DCF), a highly fluorescent molecule [9].

Assay Procedures: Treatments and Exposure

The test is carried out on human keratinocytes (HuKe), as described previously in the reference for [10]. The cell viability after UVA exposure and without UV exposure is checked by NRU (Neutral red uptake) assay [11].

The percentage of reduction of ROS release is determined according to the following formula:
Percentage of reduction = 100 − (fluorescence sample/fluorescence negative control)

Results Interpretation

A reduction of ROS release in the cells treated with the sample compared to untreated cells, especially if dose-related, is a proof of antioxidant/scavenging effect. The intensity of the effect is proportional to the percentage of inhibition.

On the basis of the results shown in Table 3 and Figure 2, the sample of active ingredient versus Placebo shows antioxidant/scavenging capability, in fact it shows an inhibition of UVA-induced ROS release in human keratinocytes.

**Table 3.** ROS Inhibition % induced by the active ingredient at different concentrations.

|  | Exposure | 150 μg/mL | 75 μg/mL | 37.5 μg/mL | 18.75 μg/mL | 9.38 μg/mL | 4.69 μg/mL | Vit C 150 μg/mL |
|---|---|---|---|---|---|---|---|---|
| ROS Inhibition % | 4′ UVA | 83.4 | 77.1 | 60.8 | 47.2 | 35.6 | 40.7 | 67.3 |
|  | 8′ UVA | 94.1 | 92.7 | 88.0 | 75.1 | 56.9 | 50.0 | 75.5 |
|  | 12′ UVA | 93.1 | 90.9 | 87.4 | 68.7 | 36.3 | 27.3 | 54.1 |
|  | 16′ UVA | 88.2 | 85.0 | 79.4 | 58.1 | 39.0 | 37.0 | 44.5 |
|  | 20′ UVA | 84.4 | 79.7 | 72.7 | 51.2 | 37.3 | 31.4 | 45.2 |

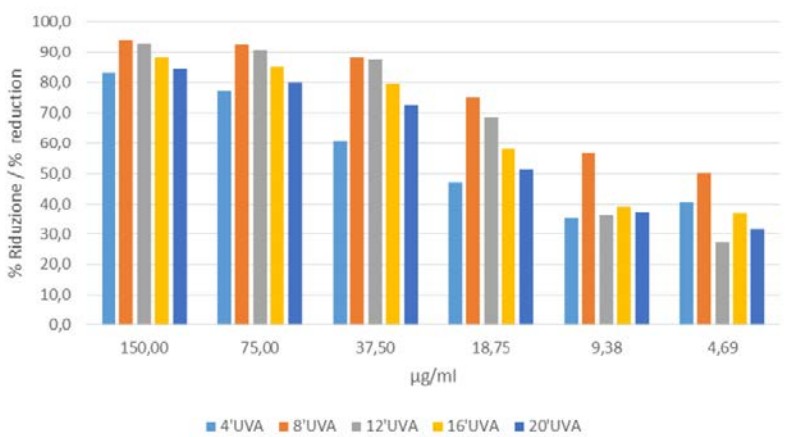

**Figure 2.** % ROS reduction.

## 4. Anti Itching Effect (Degranulation Test on Basophil Cells)

Hydrolyzed lignin 1% solution shows a slight dose dependent inhibiting effect on basophil degranulation, with a consequent anti itching effect.

Purpose of the Test

The objective of the test is to evaluate the sample's efficacy in inhibiting the degranulation of basophil cells mediated by IgE. Basophils are the terminal effectors of in vivo cells reactions. The interaction between basophils and IgEs is mediated by the high-affinity IgE receptor (FceRI). The activation of the receptor caused by antigens or aptens induces basophil degranulation, triggering an immediate hypersensitivity reaction in vivo. The ability of a tested substance or topical products to inhibit the degranulation of basophil cells may be read as an indicator of its capability to inhibit the above described symptoms and above them, mainly discussed in the reference [12].

Assay Procedures: Treatments and Exposure

The test is carried out following the IgE test method as described in the reference [13]. The active ingredient concentration was chosen after MTT cell viability assay [6]. The product was solubilized in cell culture medium at different concentrations. The product underwent a preliminary cytotoxicity screening on the cells to decide the best concentration for testing it without cytotoxic effects on the cells, in order to avoid false results. Untreated cells were used as a negative control for the spontaneous

degranulation. Sodium hyaluronan in serial dilutions was used as a control for the inhibition of basophil degranulation. A positive control consisting in cells exposed to Anti-human IgE receptor antibodies was used to evaluate the maximal degranulation level. Cell lysates in 1% Triton were also analyzed in order to evaluate the highest possible release of the analyzed molecule.

Evaluation of Degranulation

In order to evaluate the effect of the tested product on basophil degranulation, the sample was pre-incubated with a cell line of transfected rat basophil cells (RBL) expressing the IgE human receptor (FcRI) following test described in Membrane IgE Binds and Activates FceRI in an Antigen-Independent Manner [13].

The inhibition degranulation percentage of activated basophil with IgE receptor has been calculated as:

% degranulation inhibition = 100 − [(OD sample + PC − OD CN)/ (OD CP − OD CN)] × 100

PC: positive control (Anti-human igE receptor)

CN: negative control (spontaneous degranulation)

Results and Conclusions

As shown in Table 4, the results are reported being expressed as an index of basophil degranulation.

**Table 4.** % Degranulation Inhibition.

| Samples | % Degranulation Inhibition | | | |
|---|---|---|---|---|
| 1% Active Ingredient Solution | 5 μg/mL | 2.5 μg/mL | 1.25 μg/mL | 0.06 μg/mL |
| | 7.5% | 3.4% | 2.2% | 0.1% |
| Hyaluronic Acid (HA) in vitro degranulation Inhibitor | 10 mg/mL | 5 mg/mL | 2.5 mg/mL | 1.25 g/mL |
| | 79.9% | 56.3% | 45.0% | 22.6% |

The tested sample is slightly effective in inhibiting basophil degranulation in a dose dependent way. The highest effect is pointed out at the highest tested concentrations. Active Ingredient 1% solution compared to Hyaluronic acid shows a slightly higher dose dependent inhibiting effect on basophil degranulation.

## 5. Anti Pollution Effect

On the basis of the results below, a sample containing hydrolyzed lignin did show an in vitro protection against air pollutants stress on human keratinocytes.

*In Vitro Evaluation of Protective Effect of a Raw Material Against Air Pollutants on Fibroblasts*

Purpose of the Test

This study concerns the in vitro evaluation of the protective effect of cosmetic product or raw materials on murine fibroblast. The cells are stressed with a certified standard atmospheric particulate materials collected in an urban area, Detroit (NIST). This standard contains polycyclic aromatic hydrocarbons, pesticides, polychlorinated biphenyls and other organic pollutants in defined concentrations, with a particle size mainly ranging from 1 μm to 10 μm.

Particulate matter such as PM 10, PM 2.5, PM 1 is defined as the fraction of particles with an aerodynamic diameter smaller than respectively 10, 2.5, 1 μm. The protective properties are evaluated through the investigation of the cell viability by MTT assay, with and without the protecting agent. Suitable controls are enclosed. This assay can point out a protection towards organic pollutants in small size particles (PM 1, PM 2.5 and PM 10).

Assay Procedures: Treatment and Exposure

The test is carried out on a murine fibroblasts (Balb/3T3 cells, clone A31). Cells are cultured in DMEM containing 10% CS and antibiotics. The standard contains polycyclic aromatic hydrocarbons,

pesticides, polychlorinated biphenyls and other organic pollutants in defined concentrations, with a particle size mainly ranging from 1 μm to 10 μm [14].

Cells were seeded in 96 wells plates and allowed to grow for 24 h at 37 °C and 5% CO2.

The second day medium was replaced with a fresh one, supplemented with 2 dilutions of the tested product and with 2 concentrations of air pollutant stress. The sample has been dissolved directly in the culture medium. The test has been carried out in three replicates for each test dilution.

Stressed cells untreated with the sample and unstressed untreated cells have been used as controls in the experiment. At the end of the incubation period, the cells are tested for their viability with the cytotoxicity (MTT) assay.

The cell viability is expressed in percentage terms:

% cell viability = [OD(570 nm − 650 nm) test product / OD(570 nm − 650 nm) negative control] × 100

So the cell death rate is calculated as:

% cell death = 100 − % cell viability

To evaluate the viability cell protection:

% protection from pollution stress = pollutant stress

(% cell death CSNT − % cell death CST)/ % cell death CSNT × 100

Where

CSNT = Stressed cells not treated with the sample

CST = Stressed cells treated with sample

If the sample has an efficacy, a statistical analysis is reported to evaluate the statistical significance of the data.

A $p$ value < 0.05 indicates a statistically significant result.

Results Interpretation

A different cell mortality value between units treated with urban dust stress and a sample, and units treated with only urban dust is an index of protective and epidermis barrier effect of the sample.

Results and Conclusion

On the bases of the results obtained and listed in Table 5, the sample of active ingredient versus Placebo did show an in vitro protection against air pollutants stress on human keratinocytes.

**Table 5.** Results: % protection from pollutants stress.

| Active Ingredient | % Viability Cell (DS) | Cell Death % | % Protection from Pollutants Stress |
|---|---|---|---|
| 0.4 % | 102.03 (6.69) | N/A | N/A |
| 0.2% | 92.64 (8.00) | 7.36 | N/A |
| 0.4% + NIST 3mg | 17.04 (2.23) | 82.96 | No protection |
| 0.2 % + NIST 3mg | 20.77 (2.86) | 79.23 | No protection |
| 0.4% + NIST 1mg | 52.07 (5.06) | 47.93 | 19.35 |
| 0.2 % + NIST 1mg | 53.49 (1.79) | 46.51 | 22.33 |
| NIST 3mg | 23.98 (1.10) | 76.02 | N/A |
| NIST 1mg | 40.12 (3.84) | 59.88 | N/A |
| Negative control | 100.00 (2.70) | N/A | N/A |

## 6. Dermatitis

Dermatitis is a condition characterized by reddened skin, irritation, swelling and itching. The most common forms of dermatitis have an irritating or allergic origin and correspond to a reaction to external or internal factors that shows up with a widespread rash.

Dermatitis is a skin reaction to external factors (allergens, chemicals, physical factors) or internal factors (release of inflammation factors). It is characterized by a sudden inflammation of the skin,

which becomes red and itchy. It can be temporary or persistent, depending on the causes, and can be complicated by swelling, desquamation, vesicles, blisters, erosions and crusts.

The most common forms of dermatitis are, for example:

- Atopic Dermatitis: very common in childhood, it manifests with redness and blisters at the folds of the skin, e.g., in the elbows, knees and neck, where moisture promotes skin irritation. Atopic dermatitis is a chronic inflammatory skin disease associated with cutaneous hyperreactivity to environmental triggers. The clinical phenotype that characterizes atopic dermatitis is the product of interactions between susceptibility genes, the environment, defective skin barrier function, and immunologic responses [15].
- Seborrheic Dermatitis: skin inflammation is accompanied by intense desquamation. Seborrheic dermatitis affects the scalp, central face, and anterior chest. In adolescents and adults, it often presents as scalp scaling (dandruff). Seborrheic dermatitis also may cause mild to marked erythema of the nasolabial fold, often with scaling. Stress can cause flare-ups. The scales are greasy, not dry, as commonly thought. An uncommon generalized form in infants may be linked to immunodeficiencies. Topical therapy primarily consists of antifungal agents and low-potency steroids [16].
- Contact Dermatitis: caused by contact with stinging substances (such as nettle) or irritants (such as detergents or other chemicals) or insect poison, is strongly irritating and can give rise to vesicles in the affected area [17].

## 7. In Vivo Evaluation of Topical Formulations Based on a Lignine Hydrolysate

Examples of two topical formulations – the first for atopic and the second for seborrheic dermatitis - based on a lignin hydrolysate (active ingredient) are reported below for illustrative and non-limiting purposes [18]. The product has not shown any allergic or sensitizations reactions. All measurements of skin hydration by electrical methods and transepidermal water loss (TEWL) have been performed following EEMCO Guidelines [19].

Evaluation of the Efficacy of a Formulation for Atopic and very Dry Skin (Xerosis)

The efficacy of a cosmetic formulation containing a lignin hydrolysate in comparison to a placebo has been evaluated.

Protocol – Double blind study

The active product and the placebo are applied on the forearms by 20 volunteers with atopic and very dry skin (xerosis), twice a day, for 4 weeks.

The test is carried on 20 volunteers of either sexes (1 male and 19 females) with atopic and very dry skin (xerosis), average age 50.0 years.

During the study, subjects were instructed to wash their body using their current skin care regimen and not to apply the tested products on any site other than the prescribed ones. For the whole duration of the test, the subjects were discouraged from using different products on the forearms, and were encouraged to avoid UV exposure.

The study was performed as a randomized double blind study.

The side of application (left or right forearm) of the two formulations (active cream and placebo) were randomized among the volunteers. Each sample was labeled "right" or "left", indicating the side of application of the product. The assignment of subject number and subsequent placement on the randomization chart were made in order of appearance at the study center on the

first day. The products were given to the subjects in anonymous containers, which did not provide any information about the treatment.

Instrumental measurements of skin hydration and trans-epidermal water loss (TEWL) were performed on a selected area (9 cm$^2$) of the forearms at the baseline and after 4 weeks of treatment, as shown in Table 6, where we made a comparison between active and placebo cream [19].

**Table 6.** Mean values of skin hydration recorded for active cream and placebo product (t-test).

| Products | T0 | T4 Weeks | Variation (%) T4 Weeks – T0 | p-Level T0 vs T4 Weeks |
|---|---|---|---|---|
| active cream | mean 26.9 *st. dev. 1.8* | mean 45.6 *st. dev. 8.6* | + 18.7 (+ 69.5%) | p < 0.0001 |
| placebo cream | mean 26.7 *st. dev. 2.2* | mean 42.1 *st.dev. 7.7* | + 15.4 (+ 57.7%) | p < 0.0001 |

Active Cream vs Placebo cream p < 0.05.

Results

Skin Hydration

A statistically significant increase in the mean basal values of skin hydration were evidenced after 4 weeks of application of both the active and the placebo products.

A statistically significant difference between the active and placebo products was detected after 4 weeks of treatment, as shown in Figure 3.

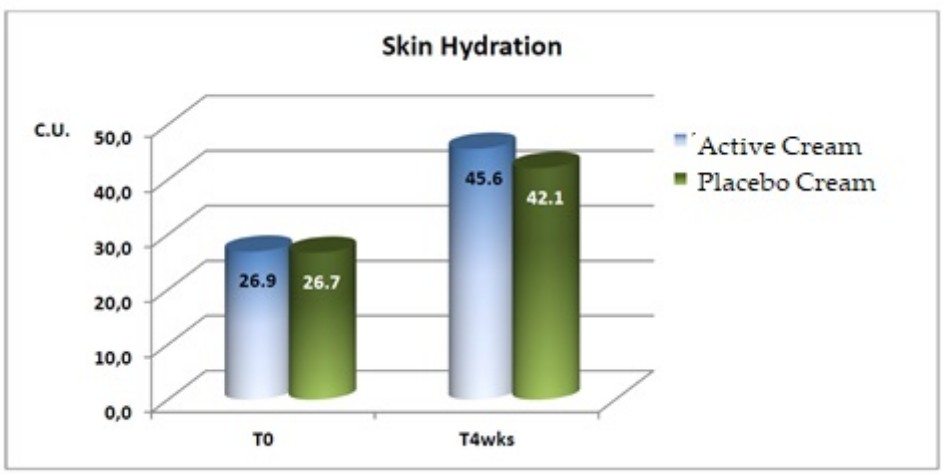

**Figure 3.** Mean values of skin hydration recorded for active cream and placebo product.

Trans-Epidermal Water Loss (TEWL)

A statistically significant decrease (improvement) in the mean basal values of transepidermal water loss was recorded after 4 weeks of application of both the active and the placebo products, as shown in Table 7 and Figure 4 [20].

**Table 7.** Mean values, standard deviations, variations and statistical significance (t-test).

| Products | T0 | T4 Weeks | Variation (%) T4 Weeks – T0 | p-Level T0 vs T4 Weeks |
|---|---|---|---|---|
| active cream | mean 9.96 *st. dev. 2.43* | mean 7.32 *st.dev. 1.19* | −2.64 (−26.5%) | p < 0.0001 |
| placebo cream | mean 10.35 *st. dev. 2.56* | mean 9.12 *st. dev. 2.00* | −1.23 (−11.9%) | p < 0.05 |

Active Cream vs Placebo Cream p < 0.0001.

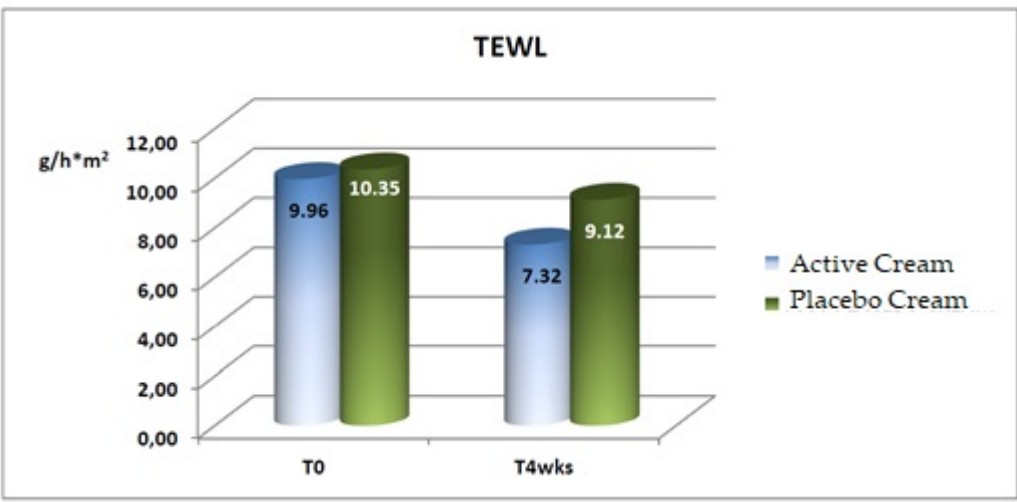

**Figure 4.** Mean values of TEWL recorded for active cream and placebo product.

Conclusions

The instrumental evaluations performed at the beginning and after 4 weeks of treatment with the formula based on a lignin hydrolysate (active ingredient) gave the following results:

- a statistically significant increase in the skin hydration (+69.5%);
- a statistically significant improvement in the trans-epidermal water loss (−26.5%).

The efficacy of the active formulation was confirmed by a statistically significant difference between active cream and placebo for both the considered parameters.

Instrumental and Clinical Evaluation of a Cream for Seborrheic Dermatitis

The efficacy of a cosmetic formulation containing a lignin hydrolysate in a face cream for seborrheic dermatitis has been evaluated.

Protocol

20 volunteers with seborrheic dermatitis on the face (6 females and 14 males, average age 50 years), were selected for the study.

They applied the product on the face on a daily basis for 4 weeks.

During the study, subjects were instructed to wash their face using their current skin care regimen and not to apply the tested product on any other site than the prescribed one. For the whole duration of the test, the subjects were discouraged from using different products on the face and or obtaining UV exposure.

Results

Instrumental Evaluation

Skin Sebum

After 4 weeks of application, a statistically significant decrease in the mean values of skin sebum was detected (Table 8 and Figure 5).

**Table 8.** Mean values, standard deviations, variations and statistical significance (t-test).

| T0 | T4 weeks | Variation T4 Weeks - T0 (%) | T0 vs T 4 Weeks p-Level |
|---|---|---|---|
| mean 130.6 std. dev. 28.1 | mean 110.7 std.dev. 31.9 | −19.9 (−15.2%) | p < 0.001 |

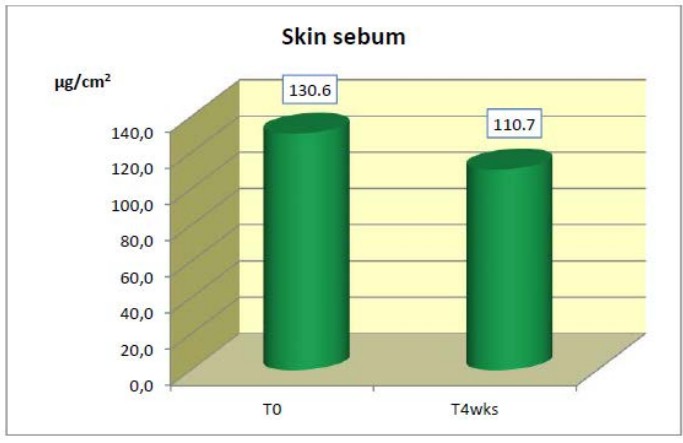

**Figure 5.** Skin sebum mean values.

Clinical Evaluation

After 4 weeks of application, statistically significant decreases in the mean values of desquamation, erythema, seborrhoea and itching were detected. The scores of desquamation, seborrhoea, erythema, itching and burning were recorded according to the 4-point scale: 0 = no reaction; 1 = mild; 2 = moderate; 3 = severe. Results are shown in Table 9.

**Table 9.** Means, standard deviations and statistical significance (Wilcoxon test).

| Symptom | T0 | T4 Weeks | p-Level |
|---|---|---|---|
| desquamation | mean 1.8 std. dev. 0.9 | mean 0.2 std. dev. 0.6 | $p < 0.001$ |
| erythema | mean 2.4 std.dev. 0.7 | mean 1.4 std.dev. 0.6 | $p < 0.001$ |
| seborrhoea | mean 2.5 std.dev. 0.5 | mean 2.1 std. dev. 0.7 | $p < 0.05$ |
| itching | mean 1.3 std.dev. 1.0 | mean 0.1 std.dev. 0.3 | $p < 0.001$ |
| burning | mean 0.3 std.dev. 0.7 | mean 0 std.dev. 0 | $p > 0.05$ |

Conclusions: The evaluations performed at the beginning and after 4 weeks of treatment, showed the following results:

- a statistically significant improvement in the mean basal values of skin sebum ($-15.2\%$),
- statistically significant improvements in the mean basal values of desquamation, erythema, seborrhea and itching.

## 8. Conclusions

According to in vitro and in vivo tests, a lignin hydrolysate compound can be used in cosmetic products as an anti aging, anti inflammatory, antioxidant, anti itching, soothing, and anti pollution agent. The compound can also be used in medical devices to help fight more painful and annoying symptoms that occur due to atopic and seborrheic dermatitis.

**Author Contributions:** Writing –review and editing (L.G.; S.U.), original draft preparation (L.G.), investigations (S.U.) data curation (S.U.), methodology (L.R.), formal analysis (L.R.), project administration (M.F.), validation (E.B.), supervision (E.B.) and conceptualization (E.B.).

**Funding:** This research was funded by Sinerga SPA, VAT NUMBER 12950420153.

**Acknowledgments:** The authors sincerely acknowledge: from Abich Elena Bocchietto for in vitro tests; ISPE for the clinical assessment.

**Conflicts of Interest:** The authors declare no conflict of interest.

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
