# Peer review of "Evaluation of the Soothing and Protective Properties of a Lignin Hydrolyzate"

_cosmetics, doi:10.3390/cosmetics6030038_

Reviewer 1 Report

The paper is interesting showing results with originality and scientific approach

Author Response

Dear reviewer, thanks a lot for your kind feedback.
Please, see updated version of our article attached.
Kind regards,
Letteria Greco

Reviewer 2 Report

How many affiliations should the paper show? I see 5 authors, are they from different institutes?

What´s about conflict of interest of the authors? Are companies involved?

Are there known allergic reactions or sensitizations to the product?

Were the physical meadurements (hydration, TEWL) performed under standardized conditions?

Author Response

1How many affiliations should the paper show? I see 5 authors, are they from different institutes?

2What´s about conflict of interest of the authors? Are companies involved?

3Are there known allergic reactions or sensitizations to the product?

4Were the physical meadurements (hydration, TEWL) performed under standardized conditions?

1- we added affiliations

2- yes, a company is involved

3- please see pag 3, raw n.278

4- yes, please, see pag 3, raw n.279

Dear reviewer, thanks a lot for your comments.
Please see above answers to your questions and attached updated version of the article.

Kind regards,

Letteria Greco

Reviewer 3 Report

‘Evaluation of the soothing and protective properties 3 of a lignin hydrolyzate’ by Greco et al seems interesting paper. Authors have a data but presented as difficult to understand. It need to be modified accordingly few facts, as follows:

hydrolysed lignin structure need to be there. Are hydrolysed lignin compositions are different for different tree source? Can you use hydrolyze word in the manuscript?

Please follow the typical paper writing style. 1. ANTI-INFLAMMATORY EFFECT…2. ANTI OXIDANT EFFECT.. 3. ANTI ITCHING EFFECT (Degranulation test on basophil cells)….etc are written in review style. It should be article style; for result section 1, 2 3...etc. you can explain background, experiment and findings with little conclusions in every section without specifying one by one for independent phenomena. Just by giving statement as a result, it look like article style. For example. 1. ANTI-INFLAMMATORY EFFECT …should be ‘hydrolyzed lignin abrogates the IL-1A production from skin cells and possess anti-inflammatory properties’.

Also finally make a combined conclusion. Don’t conclude separate conclusions in separate Tab.

Authors mentioned about molecular modeling work …I don’t see any data or exact references of molecular modeling in the current article.

How many independent replicates you have all the experiments. I don’t see standard deviations/errors or statistics in your most of result such as table 1, graph 1(with the table), Table 4, Table 5 etc.

Experimental questions:

1: Why did you pick only IL-1A. IL-1A is not as potent inflammatory as IL-1B? You should have scored IL-1B. IL-1A induced by SLS need to be backed up by literature. Does SLS induces Calpain pathway? Which skin cells you have used. If you use SLS in cell culture your cell will die, because; it’s denaturing agent. Then how can you conclude final findings. Did you use ectopic (like dermatitis model) please put the name of mouse strain. Your method is inadequate. Table need to be shown as a graph. In table 1 row 4 (Solution 0,1% SLS 0,3%) and row 6 (Solution 0,1% SLS 0,3%) is same: what you want to show? I am not able to understand. Also what is - Solution 0,1% sample of active ingredient ? Can you name the Solution - Solution 0,1% (write formulation).

2: SOD activity were performed on what cells? The sample has been solubilized in glycerin at 10%.......what samples? What is diluted.. Can you write exact name of chemical or compound. Also write the source of compound such as vendor or product with exact catalog number. If you have purified put Mass Spec/HPLC based purity data in supplementary or Figure 2. You are calling your drug as Sample is complete nonscientific. (Neutral red uptake) assay…..data need to be there in this article.

3: need to work on writing. It does not have flow. What is 1% Active Ingredient Solution… Hyaluronic Acid (HA) in vitro degranulation Inhibitor…what is that? Have you scored Bradykinin?

4: what is the source of NIST? What is the size of PM1, PM2.5 and PM10?

5: 4. DERMATITIS should be your 5. DERMATITIS. This section is your actual strength of the paper. If you re-write your section 5 very well, it will be a mile stone article.

Author Response

Dear editor, thanks a lot for your feedback.

Please see below our answers.

1hydrolysed lignin structure need to be there. Please see raw n.39 pag.1.

Are hydrolysed lignin compositions are different for different tree source? Can you use hydrolyze word in the manuscript? –> we suppose that composition is the same because they belong to the same botanical family. We introduces hydrolysed word in full text.

 2 Please follow the typical paper writing style. 1. ANTI-INFLAMMATORY EFFECT…2. ANTI OXIDANT EFFECT.. 3. ANTI ITCHING EFFECT (Degranulation test on basophil cells)….etc are written in review style. It should be article style; for result section 1, 2 3...etc. you can explain background, experiment and findings with little conclusions in every section without specifying one by one for independent phenomena. Just by giving statement as a result, it look like article style. For example. 1. ANTI-INFLAMMATORY EFFECT …should be ‘hydrolyzed lignin abrogates the IL-1A production from skin cells and possess anti-inflammatory properties’. à We tried to re-write in article style.

 3

Also finally make a combined conclusion. Don’t conclude separate conclusions in separate Tab. We did separate conclusions for each experiment, as well we kept the final conclusion.

 4

Authors mentioned about molecular modeling work …I don’t see any data or exact references of molecular modeling in the current article. Please see raw 60, pag.2.

 5

How many independent replicates you have all the experiments. I don’t see standard deviations/errors or statistics in your most of result such as table 1, graph 1(with the table), Table 4, Table 5 etc. All in vitro tests are performed in two replicates.

Were the physical meadurements (hydration, TEWL) performed under standardized conditions? Yes, as indicated at pag.9, raw 291.

 6

Experimental questions:

1: Why did you pick only IL-1A. IL-1A is not as potent inflammatory as IL-1B? You should have scored IL-1B. IL-1A induced by SLS need to be backed up by literature. Does SLS induces Calpain pathway? Which skin cells you have used. If you use SLS in cell culture your cell will die, because; it’s denaturing agent. Then how can you conclude final findings. Did you use ectopic (like dermatitis model) please put the name of mouse strain. Your method is inadequate. Table need to be shown as a graph. In table 1 row 4 (Solution 0,1% SLS 0,3%) and row 6 (Solution 0,1% SLS 0,3%) is same: what you want to show? I am not able to understand. Also what is - Solution 0,1% sample of active ingredient ? Can you name the Solution - Solution 0,1% (write formulation). We exploded deeply the test, please see pag. 2 and 3, highlighted text + edits in table 1.

2: SOD activity were performed on what cells? The sample has been solubilized in glycerin at 10%.......what samples? What is diluted.. Can you write exact name of chemical or compound. Also write the source of compound such as vendor or product with exact catalog number. If you have purified put Mass Spec/HPLC based purity data in supplementary or Figure 2. You are calling your drug as Sample is complete nonscientific. (Neutral red uptake) assay…..data need to be there in this article. As specified in pag 4, raw 123, it is an in tube enzymatic reaction. We prepared a solution of 10% of active ingredient in glycerin and we tested three different concentrations as now specified at raw 135, pag.4.

3: need to work on writing. It does not have flow. What is 1% Active Ingredient Solution… Hyaluronic Acid (HA) in vitro degranulation Inhibitor…what is that? Have you scored Bradykinin? Please, see pag.6, we tried to explain better the performed test.

4: what is the source of NIST? What is the size of PM1, PM2.5 and PM10? We detailed the composition and the unit of measure as you can see at raw 220, pag.7

5: 4. DERMATITIS should be your 5. DERMATITIS. This section is your actual strength of the paper. If you re-write your section 5 very well, it will be a mile stone article. We added details about dermatitis, as you can see through highlighted text.

I hope to have informed you well,

Kind regards,
Letteria Greco

Round  2

Reviewer 3 Report

Dear Authors next time please consider N=3 experimental replicates.

I am impressed with your manuscript; especially Dermatitis section. You deserve publication.

Keep up the good work.